# Characterization of Rare Himalayan Balsam (*Impatiens glandulifera* Royle) Honey from Croatia

**DOI:** 10.3390/foods11193025

**Published:** 2022-09-29

**Authors:** Saša Prđun, Ivana Flanjak, Lidija Svečnjak, Ljiljana Primorac, Maja Lazarus, Tatjana Orct, Dragan Bubalo, Blanka Bilić Rajs

**Affiliations:** 1Department of Fisheries, Apiculture, Wildlife Management and Special Zoology, Faculty of Agriculture, University of Zagreb, Svetošimunska Cesta 25, 10000 Zagreb, Croatia; 2Faculty of Food Technology Osijek, Josip Juraj Strossmayer University of Osijek, Franje Kuhača 18, 31000 Osijek, Croatia; 3Institute for Medical Research and Occupational Health, Ksaverska Cesta 2, P.O. Box 291, 10001 Zagreb, Croatia

**Keywords:** *Impatiens glandulifera*, honey, pollen spectrum, physicochemical properties, sensory profile

## Abstract

Himalayan balsam (*Impatiens glandulifera* Royle) is an invasive garden ornamental plant species originating from Asia, which produces significant amounts of nectar. In Croatia, it is widely distributed along the banks of the Mura River. Although this plant species is widespread in Europe, there are still no available scientific data about this unifloral honey type. The results showed that Himalayan balsam honey is characterized by the high presence of pollen grains in the pollen spectrum (59–85%), natural higher diastase activity (39.1 ± 7.98 DN), negative specific rotation (−21.2° ± 6.89) and an extra light amber color (48.5 ± 12.69 mm Pfund). The carbohydrate profile is characterized by monosaccharides fructose (39.34 ± 0.65 g/100 g) and glucose (31.91 ± 1.42 g/100 g) with a ratio >1.23, while the most commonly represented disaccharide was maltose (3.04 ± 0.79 g/100 g). The average total phenolic content was 130.97 ± 11.17 mg gallic acid/kg honey, and the average antioxidant capacity value was 225.38 ± 29.58 µM Fe(II). The major mineral element was K, with an average of 533.92 ± 139.70 mg/kg. The sensory profile was characteristic with a light orange color and medium-intensity odor and aroma. The crystallization rate was moderate and characterized by the appearance of opalescence and gelatinous forms of crystals. The results of this study provide the first insight into the melissopalynological, physico-chemical and sensory profile of Himalayan balsam honey.

## 1. Introduction

Honey is one of the greatest natural products, with a specific chemical composition and high nutritional value. The physicochemical characteristics and sensory properties of honey (nectar, honeydew or their mixtures) are used to determine its quality, and they are greatly influenced by the nectar/honeydew origin, soil type, climate conditions and post-harvest handling practices [1,2]. The determination of the physicochemical properties of honey and its quality control are crucial for beekeepers to meet the demand for the local and international market through compliance with legal honey standards and authenticity confirmation [3]. The water content in honey is largely conditioned by the technological activities of beekeepers, honey antioxidant content as well as various physicochemical properties, which are primarily dependent on the botanical origin and different geographical factors, while processing and storage conditions have minor effects on the overall composition of honey. Although there are more than 100 botanical species in Europe that can produce sufficient amounts of nectar for the production of unifloral honeys [4], only a small number of honey types have been described and characterized in terms of their uniflorality. The use of foreign ornamental plant species in gardens and parks stimulates the spread of invasive species and thus allows the emergence of plant populations that can suppress the local flora and threaten habitat diversity.

Himalayan balsam (*Impatiens glandulifera* Royle) is a tall annual invasive plant species from the Balsaminaceae family, native in the western Himalayas. The species was first introduced to England from Kashmir in 1839 [5] as a garden ornamental plant [6] but spread widely, so today, it is considered as an invasive species in 23 European countries [7]. In the Croatian flora, the genus *Impatiens* is represented by five species (*I. parviflora*, *I. noli-tangere*, *I. glandulifera*, *I. balsamina* and *I. balfourii*), of which only the species *I. noli-tangere* L. is native, while other species are allochthones [7]. The first record of Himalayan balsam in Croatia dates back in the late 1960s and early 1970s. Although it has been recorded in several locations in Croatia, nowadays, it is most widespread in the north of the country along the banks of the Mura River (Međimurje County). It represents a significant nectar source for honeybees in late summer. Himalayan balsam is characterized by large inflorescences with purple, 3–4 cm long, zygomorphic flowers blooming in August and September. It is protanderous and self-compatible, but due to its late flowering and abundant production of nectar (19 µL/24 h with 48–53% of sugar) [8,9] and pollen, it attracts a great number of honeybees [10]. Although this plant species is widespread in Europe, mainly in riparian habitats and close to rivers and waterways [11], there are still no scientific records about Himalayan balsam unifloral honey, i.e., it has not yet been described and characterized. The International Honey Commission (IHC) has made an effort to characterize 15 unifloral honeys that are the most commonly represented in Europe due to their abundance and commercial importance [12]. However, in the last decade, it has been possible to classify more and more rare honey types as unifloral in Europe (primarily due to human impact on the spread of invasive plant species), but their melissopalynological, physicochemical and organoleptic profiles have not yet been described. Consequently, this honey type is not recognized by consumers, and as such has no adequate commercial value. Moreover, due to the lack of mentioned analytical data, the differentiation of Himalayan balsam honey from multifloral and/or similar mild unifloral honeys can be challenging, which indicates a need for the detailed characterization of this honey type.

Therefore, the aim of this study was to determine the pollen spectrum, physicochemical properties and sensory profile of Himalayan balsam honey in order to obtain its characteristic profile arising from a comprehensive set of analytical data.

## 2. Materials and Methods

### 2.1. Honey Samples

In total, ten samples of Himalayan balsam honey (HBH) were collected during August 2020 (seven samples) and August 2021 (three samples) from local beekeepers in the north part of Croatia at eight micro locations along the Mura River (Figure 1). The samples were stored in glass jars and kept in the dark at 4 °C until further analyses, which were made within six months of honey extraction.

### 2.2. Melissopalynological Analysis

Honey samples (n = 10) were analyzed via qualitative melissopalynological analysis according to the method of Von der Ohe et al. [13], i.e., by counting the I. glandulifera pollen grains necessary for uniflorality confirmation (>45%). The slides for microscopic analysis were prepared according to the guidelines of the International Commission of Bee Botany [14]. The microscopic examination and counting of pollen grains (at least 500 pollen grains per each slide at a magnification of 400–1000×) was carried out on an Axio Scope A1 (Carl Zeiss, Oberkochen, Germany) light microscope attached to a digital camera, model Axiocam 208 Color (Carl Zeiss, Oberkochen, Germany) and coupled to an analysis system (ZEN 3.1 blue edition, Carl Zeiss, Oberkochen, Germany) for the morphometric examination of the studied pollen grains. Pollen grains originating from nectarless plant species were also recorded and counted, but they were excluded from the total pollen count. Only the pollen grain types with frequencies higher than 1% were considered. The identification of pollen grains was carried out using melissopalynological collection data from the literature [15,16], along with internal reference samples of pollen grains in the form of native preparations (an in-house collection of reference pollen slides from the Department of Fisheries, Apiculture, Wildlife Management and Special Zoology, University of Zagreb Faculty of Agriculture).

### 2.3. Physicochemical Analyses

Physicochemical parameters were determined in accordance with national [17,18] and international legislation [19,20] following standardized analytical methods pre-scribed by AOAC International [21] and the International Honey Commission [22].

#### 2.3.1. Moisture Content

Moisture content was determined using the refractometric method [22] on a Mettler Toledo (Columbus, OH, USA) refractometer (Refracto 30PX) at 20 °C.

#### 2.3.2. Electrical Conductivity

Electrical conductivity was measured using a Mettler Toledo (Columbus, OH, USA) conductometer (EL3) in a 20% (*w*/*v*) water solution of honey at 20 °C [22].

#### 2.3.3. Hydroxymethylfurfural

Hydroxymethylfurfural (HMF) content was determined according to the White method [22]. Absorbance measurements of honey solutions were determined at 284 nm and 336 nm. Measurements were performed in square 10 mm path length quartz cuvettes, and the results are expressed as mg/kg of honey.

#### 2.3.4. Diastase Activity

Honey diastase activity was determined with Phadebas tablets and expressed as diastase numbers (DNs). The blue water-soluble fragments formed were determined using the spectrophotometer at 620 nm. The absorbance of the solution was directly proportional to the diastatic activity of the sample. Both methods (HMF and DN) were conducted using a Shimadzu (Kyoto, Japan) double beam spectrophotometer (UV-1800) [22].

#### 2.3.5. Specific Rotation

Specific rotation was determined with an Atago (Kyoto, Japan) polarimeter (Polax 2L) [22].

#### 2.3.6. pH, Free Acidity, Lactones and Total Acidity

The pH, free acidity, lactones and total acidity were determined via the titrimetric method [22] using a Mettler Toledo (Columbus, OH, USA) pH meter (MP 220). 

#### 2.3.7. Color

Two methods were used for honey color determination. A Lovibond Honey Color-Pod (Amesbury, UK) was used for the determination of color grading based on the measurement of transmittance at 430 and 530 nm of homogenous liquid honey, and this was compared with the transmittance of pure glycerin. The results were expressed as mm in the Pfund scale. The second method used for color determination (color intensity) was the spectrophotometric method described by Beretta et al. [23], where the difference between the absorbance of a 50% (*w*/*v*) honey solution at 450 and 720 nm was determined.

#### 2.3.8. Carbohydrates

Carbohydrate analysis was carried out on a Shimadzu (Kyoto, Japan) HPLC system which consisted of an LC-20AD prominence solvent delivery module, a DGU-20A5R degassing unit, an SIL-10 AF automatic sample injector, and an RID-10A refractive index detector. The instrument was coupled with a computer equipped with LabSolutions Lite Version 5.52 software. The analytical column used for carbohydrate (fructose, glucose, sucrose, maltose, melezitose, raffinose and xylose) separation was an Agilent Technologies ZORBAX NH2 (Santa Clara, CA, USA) (4.6 × 250 mm, 5 μm particle size). The mobile phase consisted of HPLC-grade acetonitrile (J. T. Baker, Devnter, The Netherlands) and ultrapure water (70/30, *v*/*v*), while the operating conditions were: an injection volume of 10 µL, a mobile phase flow of 1 ml/min and a temperature of 30 °C. Carbohydrates were identified according to their retention times, and quantification was performed via external calibration carried out with carbohydrate standards suitable for HPLC analysis. Fructose, anhydrous glucose, sucrose, raffinose pentahydrate and melezitose hydrate were purchased from Sigma-Aldrich (St. Louis, MO, USA), while xylose and maltose monohydrate were purchased from Kemika (Zagreb, Croatia) [21].

#### 2.3.9. Total Phenolic Content (TPC)

Total phenolic content (TPC) was determined using the modified Folin–Ciocalteu method described by Beretta et al. [23]. The method is based on the reaction of phenolic compounds with the Folin–Ciocalteu reagent in an acidic medium, resulting in a colored product. The color intensity, which is proportional to the concentration of total phenolic content in the sample solution, is measured at a wavelength of 750 nm. Gallic acid (Carl Roth GmbH, Karlsruhe, Germany) was used for quantification, and the results are expressed as mg gallic acid/kg honey.

#### 2.3.10. Ferric Reducing Antioxidant Power (FRAP)

The method prescribed by Benzie and Strain [24] was used to determine the total antioxidant capacity. This is a spectrophotometric method based on the ability of antioxidants to reduce the yellow complex of ferric iron (Fe^3+^) with 2,4,6-tris(2-pyridyl)-s-triazine (TPTZ) by donating electrons in an acidic medium into the blue-colored Fe^2+^-TPTZ complex. The intensity of the resulting color is measured at 593 nm and is proportional to the reducing ability of antioxidants. TPTZ reagent was purchased from Alfa Aesar (Haverhill, MA, USA), HCl was purchased from Sigma-Aldrich, while FeCl3·6H2O, which was used as a standard for calibration curve calculation, was obtained from T.T.T. Ltd. (Sv. Nedelja, Croatia). 

#### 2.3.11. Mineral Element Analyses

Honey samples (0.7 g) were acid digested in Teflon vessels with caps in an Ultra-CLAVE IV microwave digestion system (Milestone, Sorisole, Italy) after ultrapure water (2 mL, GenPure system, TKA, Niederelbert Germany) and purified nitric acid (3 mL, 65%, Merck, Germany; duoPUR sub-boiling distillation apparatus, Milestone, Italy) were added. Arsenic (As), barium (Ba), calcium (Ca), cadmium (Cd), cobalt (Co), chromium (Cr), copper (Cu), iron (Fe), potassium (K), magnesium (Mg), manganese (Mn), molybdenum (Mo), sodium (Na), nickel (Ni), lead (Pb), selenium (Se), vanadium (V) and zinc (Zn) were quantified via inductively coupled plasma mass spectrometry (ICP-MS) (Agilent 7500cx, Agilent Technologies, Tokyo, Japan) according to a method described in detail by Tariba Lovaković et al. [25]. The standard reference material (1570a Spinach and 1573a Tomato leaves, National Institute of Standards and Technology, Gaithersburg, MD, USA) was processed in duplicate with honey samples to control for the quality of the analytical method. The obtained recoveries ranged from 94 to 104% of the certified values. The results are expressed on a wet mass basis. Data below the method detection limit (MDL) were assigned half of the MDL value for the respective element for the purpose of descriptive statistics.

### 2.4. Sensory Analysis

Sensory analysis (the assessment of visual and olfactory–gustatory characteristics) of the HBH samples collected was carried out by a panel of three educated and trained sensory assessors. Each honey sample was evaluated using descriptive grades (1–5) in accordance with the proposed methodology [26].

### 2.5. Statistical Analysis

Physicochemical measurement data were analyzed by means of descriptive statistics using Microsoft Office Excel 2018 [27]. The results regarding other parameters studied were presented as appropriate for a particular method; these included the results of the melissopalynological analysis in the form of the pollen spectrum and the results of the sensory analysis in terms of the description of visual and organoleptic (olfactory and tasting) HBH characteristics.

## 3. Results

### 3.1. Melissopalynological Analysis of Himalayan Balsam Honey

The melissopalynological analysis of HBH samples revealed the presence of *I. glandulifera* pollen grains at a percentage of >45% (ranging from 59% to 85%) in all investigated samples, which is generally required to classify honey as unifloral, if not specified differently by national legislation [17]. Thus, these pollen grains were the predominant ones. The presence of secondary (16–45%), important minor (3–15%) and minor (<3%) pollen of various plant taxa was also confirmed. Since *I. glandulifera* is an invasive plant species, the geographical origin of HBH may also have a strong impact on the total content (%) of specific pollen grains in the pollen spectrum (Figure 2). A detailed insight into the pollen spectrum of Himalayan balsam honey samples (n = 10; HBH1–HBH10) with regard to predominant (>45%), secondary (16–45%), important minor (3–15%) and minor (<3%) pollen of the determined plant taxa is provided in Appendix A.

### 3.2. Physicochemical Parameters of Himalayan Balsam Honey

The results of the physicochemical analyses of HBH samples are shown in Table 1. The water content in the analyzed HBH samples ranged from 15.8% to 20.5%, and it was 17.2 ± 1.42% on average. The determined values of electrical conductivity in the analyzed honey samples ranged from 0.27 to 0.45 mS/cm with an average of 0.37 ± 0.07 mS/cm. The average HMF content was 21.44 ± 12.79 mg/kg, ranging from 2.40 mg/kg to 45.54 mg/kg, which is more than the allowable amount (<40 mg/kg) according to the national and international legislations. The average diastase activity in the analyzed HBH samples was 39.1 ± 7.98 DN, with a minimum value of 26.8 DN and a maximum value of 52.1 DN. The pH values in the investigated honey samples ranged from 3.91 to 4.42, with an average of 4.05 ± 0.16; free acidity ranged from 23.65 mmol/kg to 44.53 mmol/kg, with an average of 33.98 ± 6.52 mmol/kg; and total acidity ranged from 23.28 mmol/kg to 51.40 mmol/kg with an average of 35.92 ± 8.47 mmol/kg. The color of HBH was classified as extra light amber to light amber, as most of the analyzed samples (8/10) came under those two color groups. Only two samples had slightly lower color (32 and 33 mm) that classified them as white honeys. In the case of >50 mm, the color of the honey was also considered as light amber. The determined color intensity (net absorbance) was in the range of 272.5 mAU to 576.5 mAU, with an average of 408.3 ± 114.80 mAU.

The dominant carbohydrates in the analyzed HBH were the monosaccharides fructose and glucose, with average values of 39.34 ± 0.65% (38.32–40.15%) and 31.91 ± 1.42% (29.56–33.86%), respectively. The most commonly represented disaccharides were maltose (3.04 ± 0.79%) and sucrose (0.08 ± 0.05%), while the trisaccharides melezitose (0.55 ± 0.43%) and raffinose (0.13 ± 0.03%) accounted for less represented/minor sugars. Xylose was not detected in the analyzed HBH samples.

The total phenolic content and antioxidant capacity (FRAP assay) are presented in Table 2. The total phenolic content varied between 117.55 mg gallic acid/kg of honey and 150.24 mg gallic acid/kg of honey with an average of 130.97 ± 11.17 mg gallic acid/kg of honey, while the antioxidant capacity (FRAP assay) was between 199.00 µM Fe(II) and 260.88 µM Fe(II), with an average of 225.38 ± 29.58 µM Fe(II).

### 3.3. Mineral Element Analyses

The results regarding the mineral element content of HBH are presented in Table 3. The most abundant element was potassium (K) with an average value of 533.92 ± 139.70 mg/kg, followed by calcium (Ca) with an average value of 43.54 ± 17.42 mg/kg and magnesium (Mg) with an average value of 17.36 ± 6.54 mg/kg. The trace minerals of zinc (Zn), iron (Fe), manganese (Mn), copper (Cu) and molybdenum (Mo) were detected in all HBH samples at low concentrations with the average values of 0.80 ± 0.66 mg/kg, 0.39 ± 0.26 mg/kg, 1.12 ± 1.61 mg/kg, 0.12 ± 0.03 mg/kg and 1.85 ± 0.48 µg/kg, respectively.

The level of trace elements which, depending on the dose, can have toxic effects (As, Ba, Cd and Pb) was also recorded. The highest level was recorded for Ba with an average value of 44.45 ± 49.29 µg/kg, followed by Cr with an average value of 4.80 ± 4.20 µg/kg, Pb with an average value of 3.52 ± 5.02 µg/kg, Cd with an average value of 0.79 ± 0.35 µg/kg and As with an average value of 0.56 ± 0.22 µg/kg.

### 3.4. Sensory Profile of Himalayan Balsam Honey

The sensory profile of HBH is presented in Table 4. In addition to the above-mentioned extra light amber color, HBH is characterized with a light orange color and medium-intensity odor/aroma. The crystallization rate is moderate and characterized by the rapid appearance of opalescence and gelatinous forms of crystals.

## 4. Discussion

The botanical origin of honey is determined with the use of melissopalynological analysis. Although this analysis may have some limitations [28], the combination of melissopalynological analysis with physicochemical and sensory characteristics can overcome these limitations and provide reliable results for all honey types. Since there is no data in the available scientific literature regarding the pollen spectrum, physicochemical parameters and sensory profile of Himalayan balsam honey, it was compared with honeys that are similar in color and electrical conductivity values, such as false indigo (*Amorpha fruticosa*), sunflower (*Helianthus annuus*), rape (*Brassica napus*), sage (*Salvia officinalis*), lime (*Tillia* sp.) and winter savory (*Satureja montana*) to gain better knowledge of its profile.

In general, all investigated honey samples in this study were classified as unifloral Himalayan balsam honey based on higher contents of pollen grain (59–85%) counts from the respective species. The group of secondary pollen (16–45%) was represented by pollen grains from *Solidago* spp., Brassicaceae and Fabaceae; the important minor (3–15%) pollen group was represented by Brassicaceae, Fabaceae, Roasaceae, *Solidago* spp., *Frangula alnus*, Asteraceae, *Fagopyrum esculentum* and *Trifolium pretense;* and the group of minor pollen (<3%) was represented by Asteraceae (*Helinathus* form), *Salix* spp., Fabaceae, Asteraceae (*Taraxacum* form), Asteraceae (*Achilea* form), *Castanea sativa*, *Centaurea* spp., *Galium* spp., Lamiaceae and *Viola tricolor.* In addition to plant species that bloom at the same time as Himalayan balsam, the pollen spectrum of this honey also includes plant species that may stay in hives in the form of bee bread from previous nectar flow such as *Salix* spp., *Castanea sativa* and some plants from the Brassicaceae family which bloom earlier in the year and thus contaminate HBH with their pollen. Since this is an invasive plant species, the geographical origin has a strong influence on the pollen spectrum of this honey type.

European legislation [20] and international standards [19] established the maximum value of moisture content of honeys at 20%. This parameter in honey largely depends on the honey harvesting process and beekeeping technology. In addition, water content may differ between honeys depending on the source of nectar and/or weather conditions in the region where the honey was produced. Therefore, values of this parameter may change from season to season and also from year to year.

Electrical conductivity is a physicochemical parameter that has been correlated with the botanical origin of honey, and it is used for honey botanical origin identification in combination with melissopalynological analysis data. The electrical conductivity of honey is directly linked with contents of its mineral salts and organic acids, and it is widely used to differentiate between honeydew honey and nectar honey and for the classification of unifloral honey. This parameter compliments different boundaries utilized in the assurance of honey’s botanical origin. According to the national and international legislations, the maximum limit value of electrical conductivity for nectar honeys is 0.80 mS/cm. HBH belongs to the group of nectar honeys with a lower value of electrical conductivity, with an average of 0.37 ± 0.07 mS/cm (range: 0.27–0.45 mS/cm), which is similar to sunflower honey [29,30,31] but higher then sage [32,33], winter savory [31,33,34] false indigo [33,35,36,37] and rape honey [30,38,39,40].

Hydroxymethylfurfural (HMF) is also technological factor and largely depends on the storing conditions and heating process (decrystallization) of honey. The distinctive characteristics of honey are due to a large number of minor components that come from the nectar and the bees themselves. Many of these substances, which provide honey its specific flavor and some of its biological activities, are thermolabile. During honey processing, heating is frequently used to decrease viscosity or to melt crystallized honey, which causes difficulties in fractioning and packaging. The hydroxymethylfurfural (HMF) and diastase activity are two of the most important parameters used to determine the freshness of honey [37], as well as the storage duration and conditions, and they tend to increase during the processing and/or aging of the product. Several factors have been reported to influence the levels of HMF and diastase activity, such as temperature and time of heating, storage conditions, pH, and floral sources. Therefore, the content of HMF provides an indication of overheating and storage in poor conditions [41]. The average value of HMF in HBH was 21.44 ± 12.79 mg/kg, which is relatively high but still below the limit of 40 mg/kg (only one sample had a higher value) prescribed by national [17,18] and international legislation [19,20]. Since this honey type has a moderately fast crystallization, the established HMF values were probably caused by heating honey during the decrystallization process. It is known that most liquid fresh honeys have absent or very low values of HMF (<1 mg/kg), but on the other hand, there are also fresh unprocessed honeys that have HMF values of 1–15 mg/kg, which was confirmed by Zhu et al. [37], Trinh et al. [41], Thrasyvoulou et al. [42] and Flanjak et al. [43]. Although the lowest measured value of HMF in HBH was 2.5 mg/kg, this type of honey has higher enzyme activity and a slightly higher value of HMF, which also may be a natural characteristic of HBH.

Diastase activity is a quality factor which indicates honey freshness and overheating. The minimal DN value according to national and international legislation for fresh and properly processed honey is ≥8, and in HBH, the average value was 39.1 ± 7.98 (Table 2), which is higher than values obtained for sunflower, rape, winter savory and sage honey [31,32,34,40,43]. DN values given in “Main European Unifloral Honeys: Descriptive Sheets” [44] for rape and sunflower honey were also lower than in HBH (26.9 and 20.8, respectively). There is a large natural variation in diastase, but considering the results obtained for HBH where DN ranged from 26.8 to 52.1, HBH seems to have natural higher diastase activity. However, heating, which affects the amount of HMF, probably also has an effect on the increased activity of diastase.

Specific rotation is a parameter used to distinguish nectar honey and honeydew honey, but it also distinguishes specific unifloral honey types [45,46]. Negative specific rotation values for nectar honey comes from the predominance of fructose in honey carbohydrate composition that has high negative specific rotation (${\left[\alpha \right]}_{D}^{20}=-92.3{}^{\circ})$ [46]. All HBH samples had negative specific rotation values which classified them as nectar honey. Values (${\left[\alpha \right]}_{D}^{20}$) ranged from −26.1° to −10.0° with an average value of −21.2° ± 6.89 (Table 1). In comparison to sunflower, rape and sage honey with average values of −17.5°, −15.1° and −16.8° [43,44,47], respectively, the obtained average result for HBH is slightly higher. 

The analyzed HBH samples had a pH of 4.05 ± 0.16 (Table 1), which is similar to sunflower, rape, sage and winter savory honey [34,43,44]. All the analyzed HBH samples had free acidity under the prescribed 50 mmol/kg with an average value of 33.98 ± 6.52 mmol/kg, while the total acidity was similar to sunflower honey [44]. Lactones values were lowest out of all other mentioned honey types.

According to the respective values of the Pfund scale (in mm), honey color can be classified into several groups: <9 mm is water white, 9–17 mm is extra white, white is 18–34 mm, extra light amber is 35–50 mm, light amber is 51–85 mm, amber is 86–114 mm and dark amber is >114 mm [48]. The average value of HBH color expressed in mm on the Pfund scale was 48.5 ± 12.69, ranging from 32.0 to 63.5 (Table 1), which classifies it as extra light amber and light amber honey. In comparison to other honey types, the color is similar to that of sage honey [49]. The net absorbance of HBH ranged between 272.5 and 576.5 mAU with an average value of 408.3 ± 114.80 mAU (Table 1). A wide range of obtained color values could be explained due to the presence of the nectar of other botanical species in different shares.

Carbohydrate content in honey mostly depends on the botanical and geographical origin, as well as the weather, storage and processing conditions [50]. The carbohydrate profile of HBH is characterized by a fructose and glucose ratio > 1.2, which indicates that honey does not have high tendency to crystalize. Fructose dominated in all samples, with an average amount of 39.34 ± 0.65 g/100 g, followed by glucose, with an average amount of 31.91 ± 1.42 g/100 g (Table 1). The fructose and glucose sum was higher than 60 g/100 g prescribed by national and international legislation, and sucrose content was under 5 g/100 g (average amount: 0.08 ± 0.05 g/100 g). The fructose amount is comparable to that in sunflower, rape, winter savory and sage honey [34,43,49,51], but the glucose content is higher in rape and sunflower honey [51], which also have a high tendency toward crystallization. Maltose was the most commonly represented disaccharide, while all other carbohydrates were present in amounts <1 g/100 g, except xylose, which was not present.

The total phenolic content of HBH honey (Table 2) was closest to the values obtained for sage honey [43,49], while all other honey types involved in the comparison in this work had higher values [52,53,54]. In addition to its nutritional value, the antioxidant activity of honey is evaluated as one of the most interesting aspects. The average FRAP value in HBH honey was 225.38 ± 29.58 µM Fe(II), which was similar to the TPC value and closest to but higher than the results for sage honey [49,51], while rape and sunflower honey had values that are higher by a few times [55,56].

Minerals are minor constituents of honey, but they play an important role in determining its quality [57,58]. The mineral composition of honey not only reflects the soil mineral content, but it also has strong botanical specificity [59]. Additionally, a correlation has been established between the mineral composition and electrical conductivity of honey [60]. Another positive correlation has been established between the mineral content and color of honey, in so far as dark honeys contain higher amounts of certain minerals when compared to pale-colored ones. Moreover, the level of minerals in honey gives an indication of its geographical origin, contamination in the surrounding area, and also an overall measure of honey purity [58]. The major elements in honey are K, Ca, Mg and Na, while the minor elements are Cd, Cr, Co, Cu, Fe, Pb, Mn and Ni [60,61,62]. The accumulation of metals may be derived from external sources such as industrial and traffic pollution, incorrect treatment and agrochemicals [63]. Potassium was the most abundant element present in all analyzed HBH samples (Table 3). The concentration of this element ranged from 337.14 mg/kg to 819.74 mg/kg. The potassium content showed a wide variation across the range in different unifloral flower honeys. Thus, the average K values in false indigo honey ranged from 103 mg/kg to 282 mg/kg [30,37,58]; in rape honey, 105–460 mg/kg [30,40,62]; and in sunflower honey, 247–439 mg/kg [30,62], which are lower values than in HBH (533.92 mg/kg), while higher values were recorded in sage honey, 769 mg/kg [64] and lime honey, 955–1574 mg/kg [58,62,64]. Calcium was the second most abundant element in HBH samples with an average content of 43.54 ± 17.42 mg/kg. Compared with the values of Ca in other honey types, it is evident that HBH had a higher content of Ca compared to false indigo honey (16–25 mg/kg) [30,37,58], but it was lower than that in rape honey (46–54 mg/kg) [30,40,62], sunflower honey (71–173 mg/kg) [30,62,64], sage honey (173 mg/kg) [64] and lime honey (45–387 mg/kg) [58,62,64]. Potentially toxic elements, such as Ba, Cr, Cd or Pb, were recorded in all HBH samples, but in low concentrations and as such cannot present any health hazard. Even an increased consumption of HBH honey will not significantly increase their weekly intake. Those elements could reflect the presence of environmental contamination, the wearing of metallic equipment or incorrect procedures used in honey preservation [65].

## 5. Conclusions

A detailed description of the melissopalynological, physicochemical and sensory characteristics of unifloral Himalayan balsam honey has been presented in this study. All investigated honey samples were classified as unifloral Himalayan balsam honey based on the higher content of pollen grains counts from the respective species. Since this is an invasive plant species, the geographical origin has a strong influence on the pollen spectrum of this honey type. The physicochemical properties revealed that this honey type has natural higher diastase activity, negative specific rotation and extra light amber color. The carbohydrate profile is characterized by the monosaccharides fructose and glucose, while the most commonly represented disaccharide was maltose. Sensory analysis revealed that the honey is characterized by medium-intensity odor and aroma, dominated by a sweet flavor. The crystallization rate is moderate and characterized by appearance of opalescence and gelatinous forms of crystals. Since this paper presents preliminary data on the physicochemical properties of Himalayan balsam honey, further research is needed with the aim of determining other physicochemical properties with modern analytical methods (HPLC, GC-MS, FTIR-ATR, etc.) in order to obtain a complete chemical profile of this rare type of honey.

## Figures and Tables

**Figure 1 foods-11-03025-f001:**
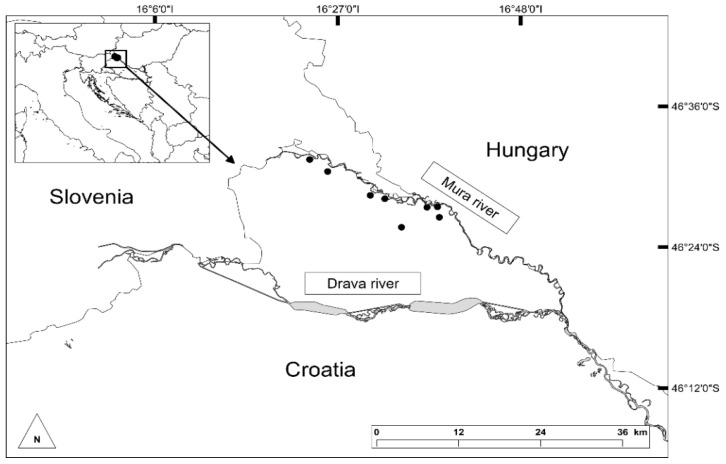
The sampling micro locations of Himalayan balsam honey along the Mura River in Croatia.

**Figure 2 foods-11-03025-f002:**
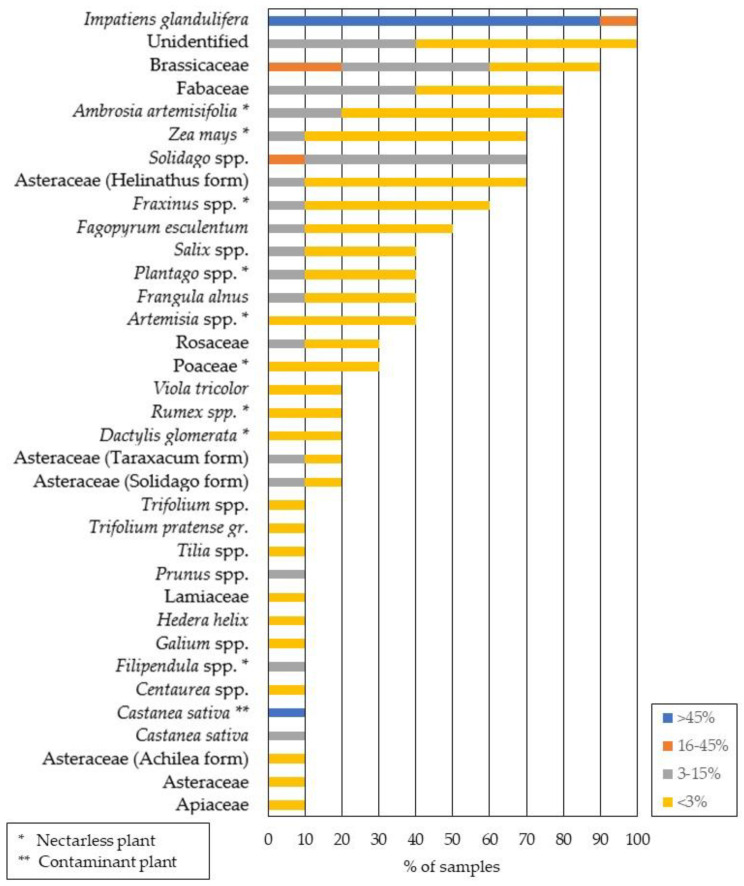
Pollen spectrum of Himalayan balsam honey (n = 10): predominant (>45%), secondary (16–45%), important minor (3–15%) and minor (<3%) pollen of plant taxa determined in analyzed honey samples.

**Table 1 foods-11-03025-t001:** Physicochemical parameters of Himalayan balsam honey.

Parameter	Unit of Measurement	Average	Minimum	Maximum	SD *
Water content	% (*w*/*w*)	17.2	15.8	20.5	1.42
Electrical conductivity	mS/cm	0.37	0.27	0.45	0.07
Diastase activity	DN	39.1	26.8	52.1	7.98
HMF	mg/kg	21.44	2.40	45.54	12.79
Specific rotation	$\alpha_{D}^{20}$	−21.2	−26.1	−10.0	6.89
Color	mm Pfund	48.5	32.0	63.5	12.69
Net absorbance	mAU	408.3	272.5	576.5	114.80
pH	-	4.05	3.91	4.42	0.16
Free acidity	mmol/kg	33.98	23.65	44.53	6.52
Total acidity	mmol/kg	35.92	23.28	51.40	8.47
Lactones	mmol/kg	2.42	0.00	6.88	2.50
Fructose	g/100 g	39.34	38.32	40.15	0.65
Glucose	g/100 g	31.91	29.56	33.86	1.42
Sucrose	g/100 g	0.08	0.05	0.23	0.05
Maltose	g/100 g	3.04	1.74	4.10	0.79
Melezitose	g/100 g	0.55	0.07	1.23	0.43
Raffinose	g/100 g	0.13	0.08	0.19	0.03
Xylose	g/100 g	ND **	ND	ND **	ND **
F + G	g/100 g	71.25	68.13	73.39	1.63
F/G	-	1.23	1.16	1.35	0.06

- values were calculated based on two replicates; * SD—standard deviation; ****** ND—not detected.

**Table 2 foods-11-03025-t002:** Total phenolic content and antioxidant capacity of Himalayan balsam honey.

Parameter	Unit of Measurement	Average	Min.	Max.	SD *
Total phenolic content	mg gallic acid/kg honey	130.97	117.55	150.24	11.17
FRAP value	µM Fe(II)	225.38	199.00	260.88	29.58

- values were calculated based on two replicates; * SD—standard deviation.

**Table 3 foods-11-03025-t003:** Mineral element content of Himalayan balsam honey.

Element	Unit of Measurement	Average	Minimum	Maximum	SD *	MDL **	<MDL ***
As	µg/kg	0.56	0.39	0.97	0.22	0.53	4/10
Ba	µg/kg	44.54	7.08	185.10	49.29	1.38	
Ca	mg/kg	43.54	18.46	63.60	17.42	1.28	
Cd	µg/kg	0.79	0.42	1.62	0.35	0.06	
Cr	µg/kg	4.80	0.33	13.66	4.20	1.08	1/10
Cu	mg/kg	0.12	0.08	0.16	0.03	0.02	
Fe	mg/kg	0.39	0.02	0.80	0.26	0.21	4/10
K	mg/kg	533.92	337.14	819.74	139.70	6.03	
Mg	mg/kg	17.36	8.57	25.73	6.54	0.46	
Mn	mg/kg	1.12	0.12	5.63	1.61	0.001	
Mo	µg/kg	1.85	1.15	2.68	0.48	0.43	
Pb	µg/kg	3.52	0.34	18.12	5.02	0.23	
Se	µg/kg	0.84	0.30	1.73	0.42	0.89	7/10
V	µg/kg	0.30	0.04	0.58	0.17	0.18	3/10
Zn	mg/kg	0.80	0.34	2.67	0.66	0.01	

- values were calculated based on two replicates; * SD—standard deviation; ** MDL —method detection limit; *** <MDL —number of samples below MDL.

**Table 4 foods-11-03025-t004:** Sensory profile of Himalayan balsam honey (n = 10).

Sensory Description	
Visual assessment	Color intensity: light to medium light
	Color tone: light amber with orange tone
Olfactory assessment	Intensity of odor: medium Description: warm, sweet, animal
Tasting assessment	Sweetness: strong
	Acidity: weak
	Bitterness: absent Intensity of aroma: weak to moderate Description of aroma: warm, candied, malt, molasses Persistence/aftertaste: short to medium
Physical characteristics	Crystallization rate: moderate

## Data Availability

Not applicable.

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
