# Peer review of "Characterization of Rare Himalayan Balsam (Impatiens glandulifera Royle) Honey from Croatia"

_foods, 2022, doi:10.3390/foods11193025_

Round 1

Reviewer 1 Report

Comments to authors are as follows:
1. Please elaborate, briefly on white method for HMF, diastase analysis and both of antioxidant analysis (TPC & FRAP).

2. Species name needs to be itallic.

3. What were the reference used for authentication and identification of pollen grains?

Author Response

Lidija Svečnjak (corresponding author)

Reviewer 2 Report

Prdun et al. have systematically characterized rare Himalayn balsam honey from Croatia for preliminary physicochemical parameters. The paper is well-written and discussed with appropriate discussion. However, the authors need to address the following comments :

1.        Abstract – the average antioxidant activity value should be expressed as microgram instead of micromolar.

2.        Abstract – a concluding statement should be included as to what the authors trying to infer from whole characterization parameters.

3.        In section 2.3, specific reference should be included in the respective characterization under each sub-section of 2.3.1-2.3.11.

4.        All the purchase details of chemicals/reagents and instruments/equipment/kits should be provided as state, city and country in the case of USA as well as city and country in the case of other countries. Also, for the second instance of same vendor/company’s mention, the authors can simply mention the company name, for instance as Sigma-Aldrich and not very time Sigma-Aldrich (USA).

5.        How do the authors justify that a panel of three educated and trained sensory assessors are sufficient? Rationalize.

6.        In section 2.5, the software and its version along with city and country details should be provided.

7.        In section 3.2, all the small one sentence paragraphs should be combined as one large paragraph although they are separate physicochemical parameters.

8.        Table 3 – why there must be asterisk in SD, MDL and <MDL, but only one asterisk reference in the footnote.

9.        The number of references should be seriously reduced to two-thirds (40-45) by removing very old references.

Author Response

Lidija Svečnjak (corresponding author)
